## Neglected Tropical Diseases

# Scoping review of knowledge, attitudes, and practices to zoonotic diseases among abattoir workers and residents in proximity to abattoirs in low-middle income countries

Oluwafemi Babatunde Daodu[1][☉], Patricia Uche Ogbo [2][☉]*, Ahmed Sherif Isa[3][☉], Toyosi Yekeen Raheem[4][☉], Uche Thecla Igbasi[5][☉], Oluwabukola Mary Ola[6][☉], Oluwatosin Eunice Olorunmoteni[7][‡], Olabisi Adejibike Oduwole[8,9][‡], Oluchukwu Perpetual Okeke[10][‡], Folahanmi Tomiwa Akinsolu[11,12][‡], Olajide Odunayo Sobande[10][‡]

**1** Department of Veterinary Microbiology, Faculty of Veterinary Medicine, University of Ilorin, Ilorin, Kwara State, Nigeria, **2** Department of Clinical Pharmacy & Biopharmacy, Faculty of Pharmacy, University of Lagos, Akoka, Lagos State, Nigeria, **3** Department of Human Physiology, Faculty of Basic Medical Sciences, Ahmadu Bello University Zaria, Kaduna, Kaduna State, Nigeria, **4** Molecular Biology and Biotechnology Department, Nigerian Institute of Medical Research, Yaba, Lagos State, Nigeria, **5** Center for Infectious Disease Research, Microbiology Department, Nigerian Institute of Medical Research, Yaba, Lagos State, Nigeria, **6** Department of Public Health, Faculty of Basic Medical and Health Sciences, Lead City University, Ibadan, Oyo State, Nigeria, **7** Department of Paediatrics and Child Health, College of Health Sciences, Obafemi Awolowo University, Ile-Ife, Osun State, Nigeria, **8** Cochrane Nigeria, Institute of Tropical Diseases Research and Prevention, University of Calabar Teaching Hospital, Calabar, Cross River State, Nigeria, **9** Faculty of Medical Laboratory Science, Achievers University, Owo, Ondo, State, Nigeria, **10** Nigerian Institute of Medical Research Foundation, Yaba, Lagos State, Nigeria, **11** Department of Public Health, Faculty of Basic Medical and Health Sciences, Lead City University, Ibadan, Oyo State, Nigeria, **12** Centre for Reproduction and Population Health Studies, Nigerian Institute of Medical Research, Yaba, Lagos State, Nigeria

☉ The authors contributed equally to this work.
‡ These authors also contributed equally to this work.
* pogbo@unilag.edu.ng, patriciaogbo@gmail.com

## Abstract

### Background

Zoonotic diseases pose a significant public health threat in low- and middle-income countries (LMICs). This scoping review aimed to map the knowledge, attitudes, and practices (KAP) of abattoir workers and residents living near abattoirs regarding zoonotic diseases in LMICs.

### Methodology/principal findings

A comprehensive search was conducted in SCOPUS, PubMed, Web of Science and Google Scholar for articles published between 2010 and 2023. Studies were included if they were conducted in LMICs, published in English, and focused on the KAP of zoonotic diseases among abattoir workers or nearby residents. Data extraction was conducted using a double-blind approach, and discrepancies were resolved by

**Data availability statement:** All relevant data are within the manuscript and its Supporting information files.

**Funding:** This work was supported by the Nigerian Institute of Medical Research Foundation [Grant Number NF-GMTP-23-082810]. The funder had no role in study design, data collection, and analysis, decision to publish or preparation of the manuscript. We declare that authors OPO and OOS received salary from Nigerian Institute of Medical Research Foundation.

**Competing interests:** The authors have declared that no competing interests exist.

consensus. Of the 4,276 articles screened, 16 met the inclusion criteria. The studies were conducted in nine LMICs, with Nigeria (6; 38%) and Ethiopia (3; 19%) accounting for the highest number. Most studies reported on knowledge (10; 63%) and practices (12; 75%), while 4 (25%) addressed attitudes. Abattoir workers showed varied knowledge of zoonotic diseases, but substantial gaps in preventive practices were observed. Notably, no studies examined the KAP of residents living near abattoirs.

## Conclusion/significance

This review underscores the need for targeted interventions to improve KAP among abattoir workers and to expand research to include nearby residents. A conceptual framework was developed to identify factors influencing KAP and guide future research, education, and policy development for zoonotic disease prevention in LMICs.

### Author summary

Zoonotic diseases, infectious diseases transmitted between animals and humans, present a significant public health concern causing an estimated 2.5 billion cases and 2.7 million deaths annually. This scoping review analysed 16 studies from nine LMICs to assess the knowledge, attitudes, and practices (KAP) of abattoir workers regarding these diseases. The findings revealed that while workers showed varying levels of knowledge, their preventive practices were often inadequate. Many did not use personal protective equipment and frequently consumed unprocessed animal products such as raw meat and milk. Importantly, no study examined the KAP of residents living near abattoirs, revealing a major research gap. We developed a conceptual framework based on the results to map the individual, cultural, and institutional factors influencing KAP. This framework can guide future studies, interventions, and policy decisions to reduce the risks of zoonotic disease transmission in LMICs.

## Introduction

Zoonotic diseases, those naturally transmitted between vertebrate animals and humans, are responsible for six out of ten infectious diseases and account for an estimated 2.5 billion human cases and approximately 2.7 million deaths annually globally [1]. Beyond this staggering morbidity and mortality, zoonosis pose a continuous threat to global health security and economic stability, with the potential to disrupt supply chains, overwhelm healthcare systems, and inflict substantial economic losses.

Transmission occurs through complex interactions among humans, animals, and the environment [1–3].

Livestock workers, such as abattoir workers, herdsmen, veterinarians, and para-veterinarian professionals, are at particularly high risk for zoonotic infections. These individuals are frequently exposed to animals or animal products and may become

infected through direct contact or indirectly via contaminated aerosols, faeces, fomites, raw meat, and unpasteurised milk [4–7]. Abattoirs, although essential for meat processing, can inadvertently serve as hotspots for zoonotic pathogens, posing risks to workers within and nearby communities [3,8].

Abattoir waste, such as animal remains and effluent, can contaminate local water sources, soil, and air, contributing to disease outbreaks [9]. Contaminated water can spread pathogens through drinking or crop irrigation [9,10]. Moreover, abattoirs attract disease vectors such as flies, ticks, and rodents, facilitating the transmission of pathogens to nearby communities [11]. These risks can be exacerbated by inadequate waste management, poor sanitation, and limited regulatory oversight, particularly in low-resource settings [12]. Consequently, abattoirs in LMICs can act as critical bridging points where animal pathogens amplify and spill over into the wider human population, making them a frontline for pandemic prevention.

The global emergence and resurgence of zoonotic diseases such as COVID-19, mpox, ebola, and lassa fever underscore the urgent need for early preparedness and prevention [13]. Critical components of preparedness include raising awareness and promoting safe practices among high-risk populations [13]. In this context, understanding the knowledge, attitudes, and practices (KAP) of abattoir workers and residents living in proximity to abattoirs is essential, particularly in low- and middle-income countries (LMICs) where both human and animal health systems are often under-resourced [14,15].

This scoping review, therefore, aims to map the existing literature on the KAP related to zoonotic infections among abattoir workers and residents living near abattoirs in LMICs. It also seeks to identify key research gaps and inform strategic interventions to mitigate the risk of zoonotic disease transmission.

Based on the findings of this review, a conceptual framework was developed to illustrate the key factors influencing KAP related to zoonotic disease prevention, which may guide future research, interventions, and policy in LMIC contexts.

## Methods

### Protocol registration

This scoping review followed the Joanna Briggs Institute (JBI) methodology for scoping reviews [16]. The protocol was registered with OSF available at https://osf.io/kb34f. This review adhered to the Preferred Reporting Items for Systematic Reviews and Meta-Analysis extension for scoping reviews (PRISMA-ScR) [17].

### Research questions

1.  What is the knowledge of abattoir workers and nearby residents in LMICs regarding zoonotic diseases?

2. What are abattoir workers' and residents' prevailing attitudes toward zoonotic diseases?

3. What preventive practices are currently adopted by abattoir workers and residents near abattoirs to mitigate zoonotic diseases?

4. What gaps exist in the knowledge, attitudes, and preventive practices related to zoonotic diseases among these populations?

### Search strategy

An initial limited search was conducted in PROSPERO, PubMed and Google Scholar on October 11, 2023 to identify relevant keywords and index terms. These were used to develop comprehensive search strategies for SCOPUS, PubMed, Web of Science and Google Scholar, targeting studies published between 2010 and 2023. The search was carried out from October 13 – 18, 2023.

The list of LMICs used was based on the Wellcome Trust's Classification [18]. The search strategy combined Medical Subject Headings (MeSH) and free text terms including 'zoonotic diseases', 'slaughterhouse', 'abattoir workers', 'knowledge', 'attitudes', 'practices' and 'proximity' (this and its synonyms did not return any results). These terms were adapted for each database and the detailed strings are provided in supporting information. We defined residents near abattoirs based on geographical proximity or distance, as our hypothesis concerns environmental exposure (air, water, noise, pathogens etc.) within the radius of 600 metres as many regulations require abattoirs to be located at least 300 metres away from residential areas to minimise health and environmental hazards.

## Inclusion and exclusion criteria

Studies were included if they were primary research studies (cross-sectional, cohort, or case-control designs), conducted in LMICs, focused on the KAP of abattoir workers and/or residents living near abattoirs regarding zoonotic diseases, and published in English.

Studies were excluded if they were systematic reviews, commentaries, editorials, conference abstracts, or research protocols, not available in full-text, not focused on KAP or zoonotic diseases, and not conducted in an LMIC.

## Selection of sources of evidence

All identified references were imported into Rayyan 2022 software, where duplicates were automatically and manually removed. To ensure consistency, all authors jointly screened the title and abstract of the first 40 articles, achieving 85% agreement.

Subsequently, two reviewers (SAI and OBD) independently screened titles and abstracts based on predefined eligibility criteria. Two reviewers (UTI and PUO) conducted full-text screening independently, with discrepancies resolved through discussion or by a third reviewer (OBD) when needed.

## Data charting process

Data extraction was carried out by two reviewers (TYR and UTI) using a pre-tested Microsoft Excel data extraction tool developed through team consensus. The tool was piloted with two articles and refined accordingly. Discrepancies during data extraction were resolved through consensus.

The following data items were extracted: first author and year of publication, country of study, study design and sample size, type of zoonotic disease(s) studied, findings on KAP, and gaps in reported KAP components.

## Synthesis of results

Results were synthesised descriptively and narratively. Extracted data were summarised narratively, thematically, and organised according to the KAP domains. Quantitative data (e.g., frequency of personal protective equipment (PPE) use, proportion with correct knowledge) were reported in frequencies, proportions, and tabulated results where appropriate consistent with the methodology of scoping review. We also analysed the gap to identify under-researched KAP domains and geographic regions.

## Critical appraisal of included studies

In line with JBI guidance for scoping reviews, no formal critical appraisal or risk of bias assessment was conducted. This review aimed to map the existing evidence, rather than to evaluate methodological quality.

## Limitations of the method

Potential limitations of this review include the exclusion of non-English language studies, which may have introduced language bias; the exclusion of grey literature may limit the evidence base's comprehensiveness; and the focus on published peer-reviewed articles may have excluded relevant programmatic or field-level reports.

## Results

### Study selection

A total of 4,404 records were identified through the systematic search. After removing 128 duplicates, 4,276 titles and abstracts were screened. Of these, 4,254 records were excluded based on the eligibility criteria. The full texts of 22 articles were retrieved and assessed, including 16 studies that met the predefined criteria [19–34]. The study selection process is illustrated in Fig 1.

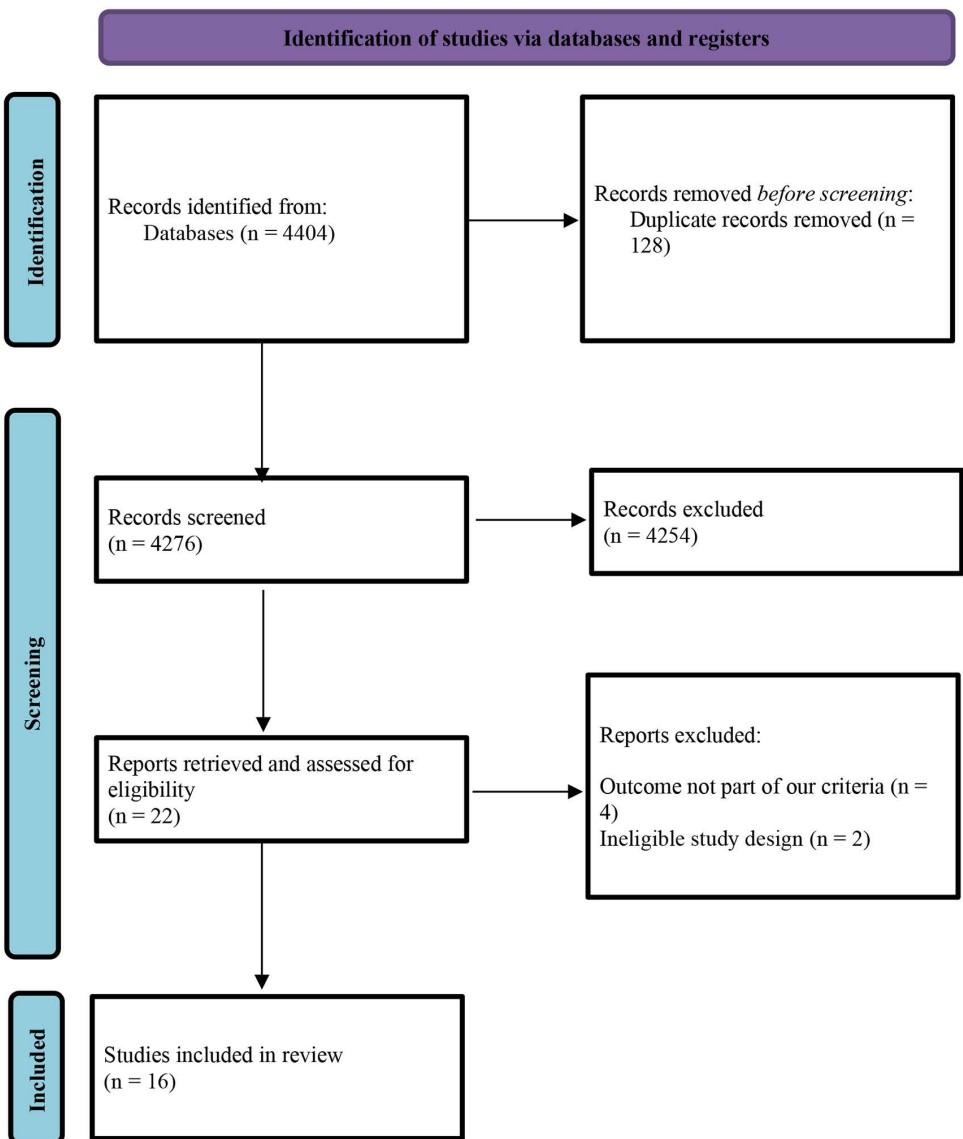

**Fig 1. Preferred Reporting Items for Systematic Reviews and Meta Analysis Extension for Scoping Reviews flow diagram [35].** The figure details how 16 studies were included from the original number of 4,404 studies found after the search. After one hundred and twenty eight (128) duplicates were removed, titles and abstracts were screened. This led to the exclusion of 4,254 studies. Full text screening was done for the remaining 22 and six more were excluded because four of the studies had outcomes different from our study criteria while the design for the remaining two studies were not eligible for our selection.

Six articles were excluded at the full-text review stage due to either an ineligible study design (n = 2) or outcomes not aligned with the review objectives (n = 4). Notably, no study examined the knowledge, attitudes, and practices related to zoonotic diseases among residents living near abattoirs.

### Article characteristics

The 16 included articles were published in 11 different journals over 12 years from 2011 to 2023. Fig 2 shows the number of articles published biennially. The most publications occurred in 2018–2019 (31%), and no article was published in 2014–2015 (Fig 2).

All included studies employed cross-sectional designs, with sample sizes ranging from 107 to 738 participants per study. Collectively, the studies involved a total of 4,178 abattoir workers, comprising 3,260 males and 918 females. The studies were conducted across eight low- and middle-income countries: five in Africa (Nigeria, Ethiopia, South Sudan, Kenya, and Cameroon), one in South America (Jamaica), and two in Asia (Myanmar and Iran). Nigeria accounted for the highest number of studies, contributing six articles (38%) to the review.

As shown in Table 1, six specific zoonotic diseases were investigated: leptospirosis (n = 1), zoonotic tuberculosis (n = 1), bovine tuberculosis (bTB) (n = 4), Crimean-Congo haemorrhagic fever (n = 1), brucellosis (n = 4), and *Toxoplasma gondii* infection (n = 1). One study addressed multiple zoonotic diseases (n = 1), while three did not specify the exact zoonotic disease investigated (n = 3).

### Knowledge

Out of the 16 included studies, 10 articles [19,21,22,25,27–29,32–34] reported on respondents' knowledge of zoonotic diseases. Six studies specifically examined knowledge related to modes of transmission [19,21,22,27,30,33], one study focused on hygiene practices [32], and four studies assessed general awareness and recognition of zoonotic diseases [22,25,28,34].

Knowledge regarding disease transmission varied significantly across studies, with reported awareness levels ranging from 31% to 93%, depending on the specific transmission route assessed. For instance, 83% of respondents knew leptospirosis could be transmitted from animals to humans [19]. In contrast, only 31% of respondents

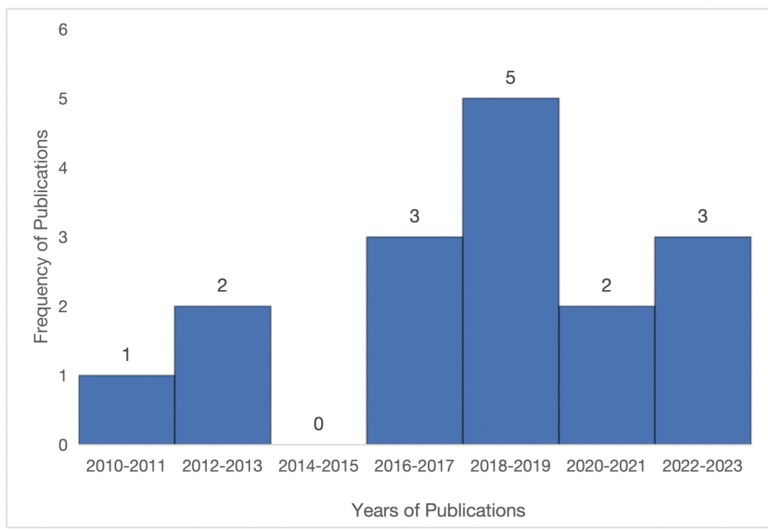

**Fig 2. Number of publications biennially.** This figure details the frequency of publications on our research topic in a two-year range.

**Table 1. Characteristics of articles selected for Scoping Review of Knowledge, Attitudes and Practices among Abattoir Workers in Low-Middle Income countries.**

| Name of 1st author and year of publication | Country of study | Study design | Zoonotic disease studied | Sample size N (M, F) |
|---|---|---|---|---|
| Brown, 2011 [19] | Jamaica | Cross-sectional | Leptospirosis | 110 (M = 74, F = 36) |
| Rosiji, 2012 [20] | Nigeria | Cross-sectional | Bovine tuberculosis | 437 (M = 315, F = 122) |
| Hambolu, 2013 [21] | Nigeria | Cross-sectional | Bovine tuberculosis | 349 (M = 273, F = 76) |
| Tsegay, 2017 [24] | Ethiopia | Cross-sectional | Nonspecific | 210 (M = 117, F = 93) |
| Cook, 2017 [22] | Kenya | Cross-sectional | Nonspecific | 738 (M = 716, F = 22) |
| Mostafavi, 2017 [23] | Iran | Cross-sectional | Crimean-Congo haemorrhagic fever disease | 190 (M = 190, F = 0) |
| Awah-Ndukum, 2018 [25] | Cameroun | Cross-sectional | Brucellosis | 107 (M = 96, F = 11) |
| Ekere, 2018 [26] | Nigeria | Cross-sectional | Brucellosis | 137 (M = 103, F = 34) |
| Fekadu, 2018 [27] | Ethiopia | Cross-sectional | Bovine tuberculosis | 300 (M = 241, F = 59) |
| Madut, 2019a [28] | South Sudan | Cross-sectional | Brucellosis | 598 (M = 326, F = 272) |
| Madut, 2019b [29] | South Sudan | Cross-sectional | Brucellosis | 234 (M = 209, F = 25) |
| Odetokun, 2020 [31] | Nigeria | Cross-sectional | Nonspecific | 203 (M = 112, F = 91) |
| Agbalaya, 2020 [30] | Nigeria | Cross-sectional | Bovine tuberculosis | 156 (M = 121, F = 35) |
| Bahiru, 2022 [32] | Ethiopia | Cross-sectional | Zoonotic tuberculosis | 113 (M = 91, F = 22) |
| Njoga, 2023 [33] | Nigeria | Cross-sectional | Bovine tuberculosis Bovine pleuropneumonia Fascioliasis Cysticercosis Liver abscess | 157 (M = 157, F = 0) |
| Sint, 2023 [34] | Myanmar | Cross-sectional | *Toxoplasma gondii* infection | 139 (M = 119, F = 20) |

M - Male; F - Female.

in other studies recognised the zoonotic transmission potential of bovine tuberculosis (bTB) or zoonotic diseases in general [21,22]. Two studies reported that 31% of the participants knew that bTB or pathogens could spread from animal to humans [21,22] while 51% of participants knew that certain food-producing animals could carry meat-borne pathogens [33]. In another study, 93% of respondents correctly identified the consumption of raw meat as a transmission route for bTB [27], while 45.6% were aware that using contaminated water during carcass processing could result in human infection [33]. Furthermore, 39.1% recognised the increased risk of zoonotic transmission due to the non-use of PPE [33].

Regarding hygiene knowledge, one study found that 73.7% of abattoir workers lacked awareness of standard hygiene practices [32]. Regarding general disease recognition, many respondents could identify zoonotic diseases when clinical signs were described in detail. However, they often failed to recognise these diseases by name, were unaware of the associated risks, or lacked basic knowledge of them [22,25,28].

## Attitudes

Three studies [19,28,31] explored the attitudes of abattoir workers toward zoonotic diseases. While respondents generally acknowledged that zoonotic transmission from animals to humans was possible, this awareness did not consistently translate into appropriate attitudes toward hygiene, ergonomics, or the use of sanitizers [31,32].

In one study, participants identified food sellers, butchers, and herders as high-risk groups due to their frequent contact with raw animal products, reflecting an understanding of occupational exposure risks [28]. Another study highlighted that while workers recognised the importance of using PPE and handwashing after handling animal body fluids, this recognition was not always reflected in their routine practices [19]. Some of the findings suggest a disconnect between

knowledge and behaviour (practice), with attitudes shaped by occupational norms, perceived invulnerability, or limited enforcement of safety standards [29,31].

### Practices

Approximately 75% of the studies reviewed reported adequate and inadequate practices within abattoirs [21–23,25,27,29,31–34]. One of the most consistent issues across studies was the limited and inconsistent use of PPE during animal handling and slaughter. A study linked poor PPE use to factors such as long-term abattoir experience, lack of knowledge, and habitual consumption of raw animal products [25]. However, one study notably reported consistent PPE use among workers [20].

Frequent direct contact with animal organs and the consumption of raw animal products such as liver and unpasteurised milk were common practices noted to significantly increase the risk of zoonotic disease transmission [25,32,34]. Although some positive hygiene practices were observed, such as regular handwashing in certain settings [34], unsafe behaviours persisted. Preventive behaviours also appeared to correlate with age and years of experience, where more experienced workers reported greater use of hygiene measures. Additionally, several studies observed inappropriate practices like reusing water to wash multiple carcasses [33], dressing carcasses on bare floors, and improper disposal of waste materials such as eviscerated foetuses [26].

Inspection practices were inconsistent across settings. While one study reported low rates of both pre- and post-slaughter inspection [31], another highlighted a combination of appropriate inspection procedures but poor waste management [20,26], reflecting variability in adherence to slaughterhouse operation and food safety standards in different settings.

Table 2 shows the overview of knowledge, attitudes and practices reported by respondents in the included articles.

## Discussion

This scoping review identified significant gaps in the KAP of abattoir workers concerning zoonotic diseases in LMICs. Importantly, no studies assessed KAP among residents living near abattoirs, indicating a substantial evidence gap in this vulnerable group. Also, only three included studies assessed attitudes as other studies did not set out with attitude assessment as an objective. Attitudes are often the least studied component of many KAP assessments because attitudes are ubiquitous, abstract, and subjective. Attitudes have affective, cognitive and behavioural components and expressing an attitude involves making a judgement about an attitude object. This complexity makes attitudes harder to define and quantify than knowledge or practices [36,37]. Knowledge and practices can be assessed using straightforward, factual or observational questions, and identified knowledge gaps and unsafe practices are easier to translate into interventions such as training, regulations, and provision of PPE. However, attitude assessments typically require longer questionnaires, Likert scale items, or qualitative methods, increasing costs and field time [37].

Among abattoir workers, knowledge about zoonotic disease transmission varied widely by disease type and geographic context. Although a general awareness of zoonotic transmission was reported in several studies, this awareness rarely translated into consistent protective behaviour. Unsafe practices, such as non-use of personal protective equipment (PPE), consumption of raw meat or milk, poor hand hygiene, and improper carcass and waste handling, were common despite some knowledge of the risks involved.

These findings suggest that awareness alone cannot change behaviour and that knowledge does not necessarily drive practices. This disconnect between knowledge and action underscores the need for targeted, context-sensitive interventions beyond awareness campaigns to address structural and behavioural barriers to safe practices.

### Factors influencing KAP

Multiple factors were identified as shaping KAP among abattoir workers. Occupational experience emerged as a double-edged sword—workers with more years in the abattoir were often more likely to use protective measures, yet their

**Table 2. Knowledge, attitudes and practices from included studies.**

| S/N | Name of 1st author and year of publication | Zoonotic Disease studied | Knowledge | Attitudes | Practices |
|---|---|---|---|---|---|
| 1. | Brown, 2011 [19] | Leptospirosis | Of the 110 respondents, 91 (83%) accurately recognised that the disease can be transmitted from animals to humans. 86 (78%) respondents had heard about leptospirosis. Of this group, 43 reported learning about the disease from a public health inspector, while 37 learned about it through print and mass media. | Out of 110 respondents, 65 (59%) believed that leptospirosis could be cured. 94 (85.5%) respondents considered using PPE (including gowns/aprons, boots, gloves, and hats) and starting each day with clean gear highly important. Nearly all respondents emphasised that hand washing is a crucial safety measure after exposure to animal body fluids in the work environment. | Of 110 workers 99 (90%) had health certification. 94 (85.5%) out of 110 respondents considered using PPE (including gowns/aprons, boots, gloves, and hats) and starting each day with clean gear highly important. |
| 2. | Rosiji, 2012 [20] | Bovine tuberculosis | Not indicated | Not indicated | Of the 450 butchers, 437 (96.6%) reported allowing veterinarians to conduct routine meat inspections on their animals. The study found good hygiene practices like hand washing and wearing protective materials like gloves and boots. The butchers reported that they employed early isolation of infected animals for treatment (29. 2%) and avoided close contact with the infected animals (23.3%) to limit cattle to cattle spread of bovine tuberculosis. More people (76; 36.7%) in the 30–39 age category and males (138; 66.7%) exhibited good measures towards bovine TB prevention. |
| 3. | Hambolu, 2013 [21] | Bovine tuberculosis | Of the 349 respondents, 294 (84.2%) had heard of bTB 107 (30.7%) knew that bTB could spread from animals to humans | Not indicated | Of the 349 respondents, 75 (21.5%) admitted consuming fuku elegushi (visibly infected parts of the lungs) before selling which predisposed them to TB. 310 (88.8%) respondents did not utilise gloves during meat processing, 50 (14.3%) consumed raw meat, 98 (28.1%) sold contaminated meat and 49 (14.0%) did not practise handwashing after processing raw meat. |
| 4. | Tsegay, 2017 [24] | Nonspecific | Not indicated | Not indicated | Of the 210 respondents, 191 (91%) respondents reported eating raw meat and 198 (94.3%) reported drinking raw milk Majority claimed using gloves and covering their mouth during slaughtering and eviscerating process. |
| 5. | Cook, 2017 [22] | Nonspecific | Of 738 slaughterhouse workers, 229 (31%) knew that animal disease could be transmitted. 310 (42%) knew that meat could be a source of disease. Only 59 (8%) of workers could name a zoonotic disease. | Not indicated | 391 (53%) respondents of the 738 slaughterhouse workers wore protective clothing. |

*(Continued)*

**Table 2.** (Continued)

| S/N | Name of 1st author and year of publication | Zoonotic Disease studied | Knowledge | Attitudes | Practices |
|---|---|---|---|---|---|
| 6. | Mostafavi, 2017 [23] | Crimean-Congo haemorrhagic fever disease | Not indicated | Not indicated | Of the 190 butchers, 49 (25.8%) had a history of cutting their hands or other body at least once during the last year.<br>143 (75.3%) workers had a history of being splashed with fluids of animal viscera for more than once on their faces and 152 (80%) on other parts of their bodies<br>Most participants (159; 83.6%) had never applied chemical disinfectant to disinfect their knives, hands and faces.<br>75 (39.7%) did not use any personal protective equipment such as masks, gloves, overalls or boots), |
| 7. | Awah-Ndukum, 2018 [25] | Brucellosis | Not indicated | Not indicated | Of 107 abattoir personnel, 59 (55%) had regular contact with animals outside the abattoir and at home.<br>49 (45.8%) regularly consumed unpasteurised milk and 23 (21.5%) manipulated aborted foetuses and other uterine contents without using personal protective equipment such as gloves.<br>All (13 out of 107; 12.1%) Brucella IgG seropositive respondents did not use PPE such as gloves during work.<br>Despite suffering previous miscarriages, two Brucella IgG seropositive pregnant women in the study were still involved in the abattoir activities.<br>Longevity in the abattoir environment, activity at the abattoir, exposure to animals outside abattoir and home environments, consumption of raw milk and lack of knowledge about brucellosis were the potential factors of non-use of PPE. |
| 8. | Ekere, 2018 [26] | Brucellosis | Of 137 slaughterhouse workers, 93 (67.9%) had not heard of brucellosis. | Not indicated | There was massive non-use of PPE during slaughterhouse operations by 97 (70.8%) slaughterhouse workers out of 137<br>89 (64.9%) of slaughterhouse workers disposed of slaughterhouse wastes, including eviscerated foetuses and pregnant uterine contents, by open-air dump method.<br>There was an illegal sale of eviscerated foetuses for human consumption by 82 (59.9%) respondents or preparation of dog food by 98 (71.5%) respondents |
| 9. | Fekadu, 2018 [27] | Bovine tuberculosis | 286 (95.3%) of 300 respondents knew about TB transmission from animals to humans.<br>221 (73.7%) knew that health-looking meat can contain TB pathogens.<br>267 (89%) knew that consumption of contaminated meat is a source of bTB infection in humans. | Not indicated | Most participants (292/300; 97.3%) did not wear PPE during work hours, 75% (225/300) of the participants were found to have eaten or consumed raw meat. |

*(Continued)*

**Table 2.** (Continued)

| S/N | Name of 1st author and year of publication | Zoonotic Disease studied | Knowledge | Attitudes | Practices |
|---|---|---|---|---|---|
| 10. | Madut, 2019a [28] | Brucellosis | From 8 focus group discussions of 6–8 members; Most participants recognised the disease once clinical signs were extensively explained. Once the disease was fully discussed, respondents gave the local names for the condition. This highlighted that they knew the disease, but possibly not the risk it posed to them. Female respondents, regardless of risk group were more aware of the zoonotic diseases compared to their male counterparts. | The views from focus group discussions with farmers revealed that they believed food sellers, butchers and herders were in more contact with raw meat than anybody else in the community. | There were basic norms taught to every generation in most of these communities. The norms and practices inadvertently protected people against infections, including zoonotic diseases. Women were excluded from information dissemination activities such as local meetings, health intervention and planning. |
| 11. | Madut, 2019b [29] | Brucellosis | Of 234 respondents, only 83 (35.5% had knowledge about zoonotic diseases | Not indicated | The majority, 189 out of 234 (84.6%) participants consumed raw animal products. 74 (31.9%) used PPE during work. 157 (67.1%) washed hands after work. |
| 12. | Odetokun, 2020 [31] | Nonspecific | Of 178 who responded to the question on zoonosis, the majority knew that slaughterhouse workers can get diseases from animals (129; 72.5%)), from the slaughterhouse environment (151; 84.8%) and meat (135;75.8%). | Although the majority of the respondents (> 70%) knew that slaughterhouse workers can get diseases from animals, from the slaughterhouse environment and meat, the effect of this knowledge was not seen in the practice of hygiene and work-related ergonomics | 58 (32.6%) of 203 respondents reported regular usage of PPE, with workers demonstrating poor personal hygiene. Of 203 respondents, 69% and 3% of respondents reported that veterinarians performed an inspection after slaughter and before slaughter, respectively. |
| 13. | Agbalaya, 2020 [30] | Bovine tuberculosis | Of the 156 respondents, those who did not know the mode of bTB transmission were 2.0 times more at risk of exposure than those with the requisite knowledge. | Not indicated | Of the 156 respondents, those who had spent more than 6 years in livestock handling were associated with 3.1 times increased risk of exposure to bTB infection. Sleeping in animal sheds was associated with an increased risk of bTB compared to those who did not. |
| 14. | Bahiru, 2022 [32] | Zoonotic tuberculosis | 73 (64.6%) out of 113 respondents did not know the main hygiene practices in the abattoir. | Not indicated | 105 (92.9%) of 113 respondents had the habit of consuming raw milk/meat. 26 (23%) had obtained training on prevention of zoonosis and personal protection. None of the abattoir workers spontaneously responded to using sanitisers as a hygiene practice. |

*(Continued)*

**Table 2.** (Continued)

| S/N | Name of 1st author and year of publication | Zoonotic Disease studied | Knowledge | Attitudes | Practices |
|---|---|---|---|---|---|
| 15. | Njoga, 2023 [33] | Bovine tuberculosis Bovine pleuropneu-monia Fascioliasis Cysticercosis Liver abscess | 81 of 157 (51.6%) respondents knew some food-producing animals can harbour meat-borne pathogens; 84 (53.5%) knew that non-use of PPE can enhance the transmission of zoonotic pathogens among slaughter-house workers. 97 (61.8%) knew that human infections with zoonotic meat-borne pathogens could result from the use of con-taminated water for carcass/meat processing during slaughterhouse operations 119 (75.8%) knew that eating or drinking while processing carcass espe-cially with unwashed hands may increase the chance of infections with zoonotic pathogens. | Not indicated | 82 (52.2%) out of 157 respondents used the same water to wash more than one carcass. 112 (71.3%) dressed carcasses on bare slaughterhouse floors. 44 (28%) respondents used PPE while processing carcass. |
| 16. | Sint, 2023 [34] | *Toxoplasma gondii* infection | 75 (54%) of 139 respon-dents had high knowledge level regarding *T. gondii* infection | Not indicated | 27 (19.4%) of 139 respondents ate raw meat. More than half (59.3%) of the slaughterhouse workers did not wear PPE when slaughtering animals. 123 (88.5%) came in contact with animal organs, mus-cles or blood 101 (85.6%) washed hand before and after eating. |

prolonged exposure also increased their risk of infection, particularly for diseases like bovine tuberculosis [30]. In some cases, more experienced workers had developed habitual practices, both protective and risky, shaped by long-term work-place norms.

Access to health information and training was another key determinant. Workers with limited exposure to health education programmes or zoonotic disease training were more likely to engage in unsafe practices [19,35]. However, even among those who had received some training, behaviour change was often incomplete, suggesting that knowledge transfer alone is insufficient without reinforcement and supportive environments.

Cultural and social norms were also suggested to have played a critical role as observed in the primary studies. Deeply ingrained behaviours, such as eating raw meat, drinking unpasteurised milk, or even sleeping in animal sheds, were considered traditional practices, not risky behaviours [29]. These findings highlight the importance of culturally sensitive interventions that engage communities to reframe traditional norms in the context of public health.

## Institutional and structural barriers

Beyond individual behaviours, systemic and infrastructural deficiencies within abattoir settings further limit the ability of workers to adhere to safe practices. Many reviewed studies reported inadequate access to PPE, poor sanitation

infrastructure, and the absence of formal hygiene protocols [14,26,29,32]. In environments where PPE is unavailable or unaffordable, expecting consistent use is unrealistic.

Meat inspection practices, pre- and post-slaughter, were inconsistently implemented, leaving gaps in detecting diseased animals before human exposure [31]. Additionally, improper disposal of waste and carcasses due to weak regulatory enforcement created environmental hazards and increased the potential for zoonotic spillover events. These structural issues call for institutional reforms and policy enforcement mechanisms that make safe practices the norm, not the exception.

### Conceptual framework

We developed a conceptual framework based on the review findings (See Fig 3) to illustrate the complex interplay between individual-level factors (knowledge, attitudes, experiences), cultural influences, and institutional enablers or constraints. This framework offers a multi-level perspective on how zoonotic disease prevention behaviour is shaped in abattoir contexts.

The framework can guide future research on zoonotic disease prevention, designing multi-pronged interventions that align with real-world constraints and informing policymakers and programme designers on areas where targeted investments (e.g., infrastructure, training, behavioural change communication) are most likely to yield impact.

### Public health and policy implications

Abattoir workers are frontline actors in the food safety chain. Their actions directly influence the risk of zoonotic transmission, not just to themselves but to consumers and the broader community. Yet, this review underscores that current practices are suboptimal, with potentially serious public health implications.

There is an urgent need to implement mandatory health and safety training, enforce standard operating procedures (SOPs) for abattoir hygiene, and ensure routine availability of PPE. Additionally, governments should invest in pre- and post-slaughter meat inspections and explore behavioural nudges and incentive-based compliance strategies.

One promising approach is periodic health screening for zoonotic infections among abattoir workers. For example, a study in Ghana reported that within six months, two previously negative workers tested positive for Crimean-Congo haemorrhagic fever, underscoring the value of biannual surveillance as both a prevention and monitoring tool [38].

### Strengths and limitations

This scoping review used a rigorous, transparent methodology, including protocol registration, comprehensive database searching, and independent, consensus-based screening and data extraction. These strengths enhance the reliability of our findings.

However, several limitations must be acknowledged. First, excluding non-English studies and grey literature may have led to omitting relevant research. Second, the included studies represent only nine of the 136 LMICs, limiting generalisability. Third, the cross-sectional design of all included studies restricts causal inference regarding KAP determinants. Fourthly, considering we conducted a scoping review, we did not carry out a formal quality of appraisal of included studies. This may restrict the interpretability of our findings.

### Recommendations

Future research should explore the KAP of residents living near abattoirs, a group currently overlooked. Studies should assess the effectiveness of culturally appropriate interventions that address individual behaviours such as not washing hands after processing raw meat, and consumption of raw meat; and systemic barriers such as open-air dump method of abattoir wastes. Applying implementation science frameworks can improve the adoption of safe practices. Finally, longitudinal and mixed-methods studies are needed to better understand behaviour change and its drivers in LMIC contexts.

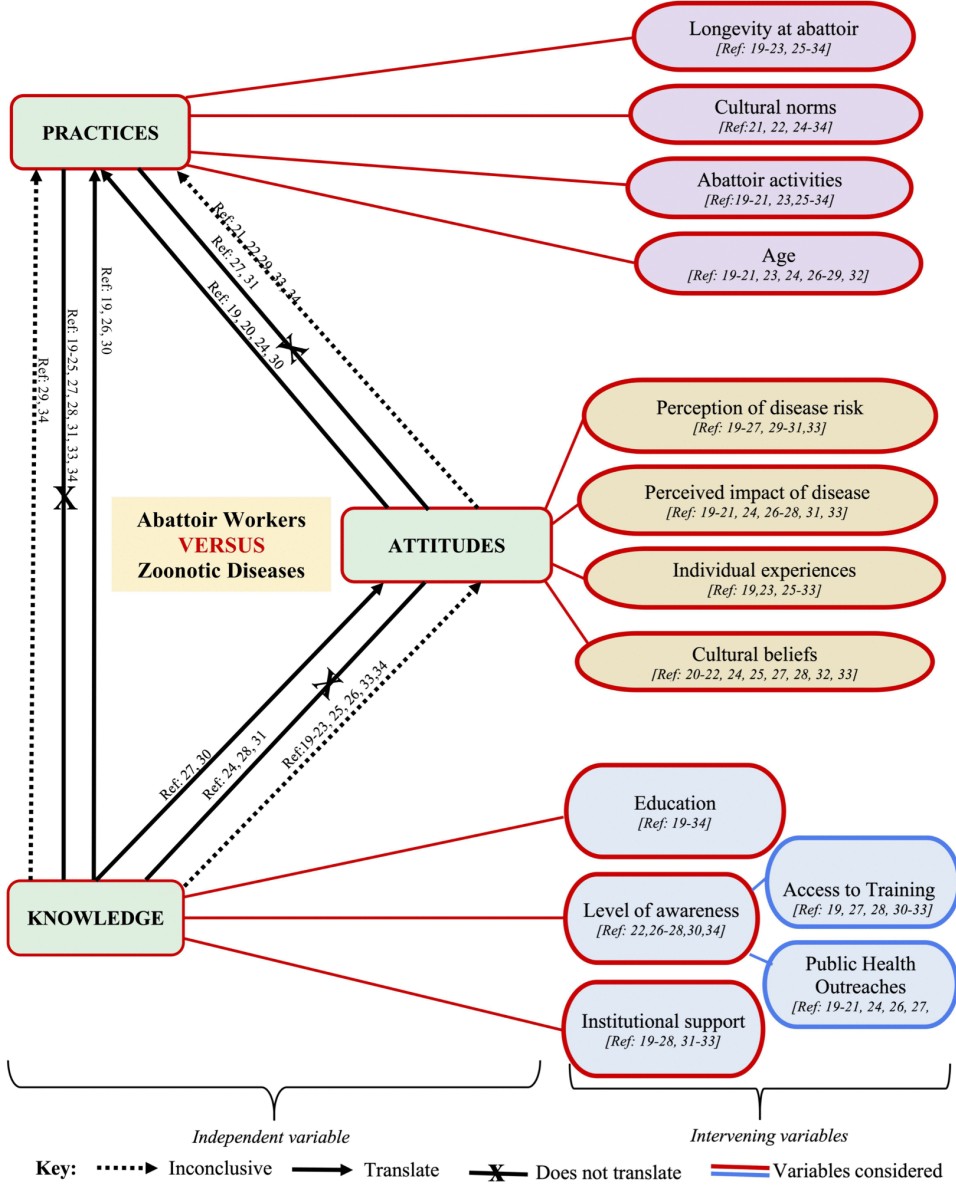

**Fig 3. Conceptual framework.** The conceptual framework shows the interplay between individual-level factors (knowledge, attitudes and practices), cultural influences, and institutional enablers or constraints. The framework offers a multi-level perspective on how zoonotic disease prevention behaviours is shaped in abattoir contexts. It also shows how knowledge, attitudes and practices relate to one another depicted as 'inclusive', 'translate' or 'does not translate'.

## Conclusion

The review showed that gaps existed in literature on studies on KAP related to zoonotic infections among abattoir workers and residents living near slaughterhouses in LMICs. This underscores the need for targeted interventions to improve KAP among abattoir workers and to expand research to include nearby residents in order to mitigate the risk of zoonotic disease transmission. Based on the findings of this review, a conceptual framework was developed to illustrate the key factors influencing KAP related to zoonotic disease prevention, which may guide future research, interventions, and policy

in LMICs contexts. The conceptual framework developed from this review offers a useful foundation for guiding future studies on KAP related to zoonotic infections in LMICs. The review also highlights the urgent need for multi-faceted interventions based on the identified knowledge gaps, modify risk-related attitudes, and reinforce the need for safe practices through institutional support and culturally sensitive health education.

Given the limited geographic coverage of existing research and the clear influence of cultural norms on abattoir practices, there is a strong need for studies across more diverse settings. This will ensure that interventions are contextually relevant and effective. The framework can also inform policy recommendations and guide the development of targeted health programmes aimed at preventing zoonotic infections among abattoir workers.

## Supporting information

**S1 Appendix. This list has the words that were abbreviated in some parts of the manuscript.** (DOCX)

**S1 Text. Search strategy for PubMed.** The search strategy presents the search terms and strings used to search PubMed. The strategy was adapted to suit other databases. (DOCX)

**S1 Table. Study characteristics.** The table of study characteristics presents the information extracted from the 16 included articles. (DOCX)

**S2 Table. This table shows all studies found from the databases.** It captures the reason for the exclusion of six studies. (DOCX)

**S1 Data. Data used to build Fig 2: The file shows the data used to build Fig 2. It contains the number of years (classified in two years period) and the number of publications during that period.** (XLSX)

## Acknowledgments

We acknowledge the Nigerian Institute of Medical Research (NIMR) Foundation for the training we received in learning how to do research. We also gratefully acknowledge Professor Morenike Folayan-Ukpong for her invaluable contribution to the development of this scoping review manuscript. Her expertise and mentorship were instrumental to the successful completion of this review.

## Author contributions

**Conceptualization:** Oluwafemi Babatunde Daodu, Patricia Uche Ogbo, Ahmed Sherif Isa, Toyosi Yekeen Raheem, Uche Thecla Igbasi, Oluwabukola Mary Ola.

**Data curation:** Oluwafemi Babatunde Daodu, Patricia Uche Ogbo, Ahmed Sherif Isa, Toyosi Yekeen Raheem, Uche Thecla Igbasi, Oluwabukola Mary Ola, Folahanmi Tomiwa Akinsolu.

**Formal analysis:** Oluwafemi Babatunde Daodu, Patricia Uche Ogbo, Ahmed Sherif Isa, Toyosi Yekeen Raheem, Uche Thecla Igbasi, Oluwabukola Mary Ola.

**Funding acquisition:** Olajide Odunayo Sobande.

**Methodology:** Oluwafemi Babatunde Daodu, Patricia Uche Ogbo, Ahmed Sherif Isa, Toyosi Yekeen Raheem, Uche Thecla Igbasi, Oluwabukola Mary Ola, Oluwatosin Eunice Olorunmoteni, Olabisi Adejibike Oduwole, Folahanmi Tomiwa Akinsolu.

**Project administration:** Oluchukwu Perpetual Okeke, Olajide Odunayo Sobande.

**Resources:** Oluchukwu Perpetual Okeke, Olajide Odunayo Sobande.

**Supervision:** Oluwatosin Eunice Olorunmoteni, Olabisi Adejibike Oduwole, Folahanmi Tomiwa Akinsolu, Olajide Odunayo Sobande.

**Validation:** Oluwatosin Eunice Olorunmoteni, Olabisi Adejibike Oduwole, Folahanmi Tomiwa Akinsolu.

**Writing – original draft:** Oluwafemi Babatunde Daodu, Patricia Uche Ogbo, Ahmed Sherif Isa, Toyosi Yekeen Raheem, Uche Thecla Igbasi, Oluwabukola Mary Ola.

**Writing – review & editing:** Oluwafemi Babatunde Daodu, Patricia Uche Ogbo, Ahmed Sherif Isa, Toyosi Yekeen Raheem, Uche Thecla Igbasi, Oluwabukola Mary Ola, Oluwatosin Eunice Olorunmoteni, Olabisi Adejibike Oduwole, Oluchukwu Perpetual Okeke, Folahanmi Tomiwa Akinsolu, Olajide Odunayo Sobande.

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
