## [Decision Letter · Decision Letter 0]

29 Jul 2025

PNTD-D-25-00848

Scoping review of knowledge, attitude, and practice to zoonotic diseases among abattoir workers and residents in proximity to abattoirs in low-middle income countries

Dear Dr. Ogbo,

Thank you for submitting your manuscript to PLOS Neglected Tropical Diseases. After careful consideration, we feel that it has merit but does not fully meet PLOS Neglected Tropical Diseases's publication criteria as it currently stands. Therefore, we invite you to submit a revised version of the manuscript that addresses the points raised during the review process.

Please submit your revised manuscript within 60 days Sep 27 2025 11:59PM. If you will need more time than this to complete your revisions, please reply to this message or contact the journal office at plosntds@plos.org. Please include the following items when submitting your revised manuscript:

We look forward to receiving your revised manuscript.

Kind regards,

Pikka Jokelainen, DVM, MPG, PhD

Academic Editor

Victoria Brookes

Section Editor

Shaden Kamhawi

co-Editor-in-Chief

Paul Brindley

co-Editor-in-Chief

**Additional Editor Comments:**

Interesting topic.

Major comment:

All conclusions must be grounded in the data, and be aligned with the study design (both of this study, and of the original studies).

Minor but important editorial comment:

Please carefully check the use of capital first letters and singular vs. plural.

**Journal Requirements:**

4) We note that your Data Availability Statement is currently as follows: "N/A". Please confirm at this time whether or not your submission contains all raw data required to replicate the results of your study. Authors must share the “minimal data set” for their submission. PLOS defines the minimal data set to consist of the data required to replicate all study findings reported in the article, as well as related metadata and methods (https://journals.plos.org/plosone/s/data-availability#loc-minimal-data-set-definition).

2) If any authors received a salary from any of your funders, please state which authors and which funders..

6) As required by our policy on Data Availability, please ensure your manuscript or supplementary information includes the following:

**Reviewers' Comments:**

Reviewer's Responses to Questions

**Key Review Criteria Required for Acceptance?**

**Methods**

-Are the objectives of the study clearly articulated with a clear testable hypothesis stated?

-Is the study design appropriate to address the stated objectives?

-Is the population clearly described and appropriate for the hypothesis being tested?

-Is the sample size sufficient to ensure adequate power to address the hypothesis being tested?

-Were correct statistical analysis used to support conclusions?

-Are there concerns about ethical or regulatory requirements being met?

Reviewer #1: In the Methods section, the authors fail to define ‘residents near slaughterhouses’ operationally, leaving the notion of ‘proximity’ unclear—whether it refers to geographic distance, duration of exposure, or functional interaction. While the authors indicate adherence to JBI guidelines for scoping reviews, which do not mandate formal quality appraisal, omitting any assessment of study quality restricts the interpretability of the findings.

Reviewer #2: (No Response)

Reviewer #3: the methodology of the papers selections is not clear. Which were the key words used to select the papers? there´s no statistics once it´s a descritive study. Ethical approval was referred.

**Results**

-Does the analysis presented match the analysis plan?

-Are the results clearly and completely presented?

-Are the figures (Tables, Images) of sufficient quality for clarity?

Reviewer #1: In the knowledge-related findings (lines 226–239), the authors report percentages without clearly linking them to the corresponding studies. The section on practices (lines 252–273) presents issues as well, combining data on PPE use, raw product consumption, and hygiene without differentiating the relevance of each practice across various zoonotic diseases.

Although the discussion is broad in scope, some interpretations exceed the evidence presented. For example, the claim of a “disconnect between knowledge and behaviour” (lines 287–288) lacks support from statistical analyses, as most studies evaluated either knowledge or behaviour but not both simultaneously. Similarly, the discussion on cultural factors and social norms (lines 302–305) relies heavily on the interpretations of primary study authors, without critically examining how these factors were measured or whether such interpretations are consistent across the nine culturally diverse countries included in the review.

Reviewer #2: (No Response)

Reviewer #3: the limitations of the study are the low numbers of papers selected. The results and tables are clearly presented

**Conclusions**

-Are the conclusions supported by the data presented?

-Are the limitations of analysis clearly described?

-Do the authors discuss how these data can be helpful to advance our understanding of the topic under study?

-Is public health relevance addressed?

Reviewer #1: Figure 3 (line 317), which presents the conceptual framework, provides a useful visual tool but has important limitations. The proposed causal relationships lack empirical support from the included studies, many of which did not test such associations. In addition, the framework does not adequately integrate structural or policy-level determinants, despite their identification in the discussion as key influences on worker practices.

The synthesis section needs a more critical reflection on the limitations of the primary studies. For example, the overrepresentation of studies from Nigeria and Ethiopia raises concerns about the generalizability of the findings to other low- and middle-income settings. Moreover, while the absence of studies on residents near slaughterhouses is noted, the authors do not fully explore the implications of this gap for public health policy.

Reviewer #2: recommendations need to be more specified and showing examples of the intervention and the address the urgent behaviors that need to be improved.

Reviewer #3: The conclusions are easily discussed once they is already expected for anyone who works in the field. It´s not possible to evaluate the conclusions without the keywords used. Maybe the number of studies could be increased if new keywords are included.

The public health relevance is addressed.

**Editorial and Data Presentation Modifications?**

Reviewer #1: Major Revision.

Reviewer #2: * regarding the title:" and residents in proximity to abattoirs" it is better to be removed as there is no publications assessed KAP among this group.

* Regarding the introduction: it needs more declaration of the effect of zoonotic diseases on public health.

* Methods: The link shown in line 101 is not correct it needs to be modified.

* line 115: the publication are from 2010 not 2000.

* line 124: comma after if they were should be removed.

*line 202: sample size should be from 107 not 110.

*line 253: what does SH refer to?

** Figure 2 why 2024 is included while the studies are till 2023 and it should be titled number of publications from 2010 to 2023.

*Table 2 is too long it is better to be divided into two tables.

* In the discussion section it needs to be mentioned why attitude is assessed in only 3 papers.

Reviewer #3: The inclusion of key words is crucial for the study to accurately evaluate its conclusions. The authors would consider including other keywords to increase the number of studies.

**Summary and General Comments**

Reviewer #1: To enhance the manuscript, the authors should expand the methodology to clarify how study limitations informed the synthesis; reorganize the results to enable clearer cross-study comparisons; revise the conceptual framework to align more closely with the available evidence; and deepen the discussion of research and policy implications, especially regarding underserved populations in low- and middle-income countries.

Reviewer #2: It is better to add list of abbreviations and avoid mentioning the abbreviation without the full name in the first time.

Reviewer #3: The study presents the results of a systematic analysis and a framework constructed based on them. Although the results were already expected, they´re fundamental if specifically shown to the government for use in the human health care benefits

PLOS authors have the option to publish the peer review history of their article (what does this mean? ). If published, this will include your full peer review and any attached files.

**Do you want your identity to be public for this peer review?** For information about this choice, including consent withdrawal, please see our Privacy Policy .

Reviewer #1: **Yes:** ANDRE PERES BARBOSA DE CASTRO

Reviewer #2: No

Reviewer #3: No

**Figure resubmission:**
---

## [Decision Letter · Decision Letter 1]

12 Nov 2025

PNTD-D-25-00848R1 Scoping review of knowledge, attitude, and practice to zoonotic diseases among abattoir workers and residents in proximity to abattoirs in low-middle income countries PLOS Neglected Tropical Diseases Dear Dr. Patricia Uche Ogbo Thank you for submitting your manuscript to PLOS Neglected Tropical Diseases. After careful consideration, we feel that it has merit but does not fully meet PLOS Neglected Tropical Diseases's publication criteria as it currently stands. Therefore, we invite you to submit a revised version of the manuscript that addresses the points raised during the review process. Please submit your revised manuscript by 25th November 2025 If you will need more time than this to complete your revisions, please reply to this message or contact the journal office at plosntds@plos.org.  Please include the following items when submitting your revised manuscript: * A rebuttal letter that responds to each point raised by the editor and reviewer(s). You should upload this letter as a separate file labeled 'Response to Reviewers '. This file does not need to include responses to any formatting updates and technical items listed in the 'Journal Requirements' section below. * A marked-up copy of your manuscript that highlights changes made to the original version. You should upload this as a separate file labeled 'Revised Manuscript with Track Changes '. * An unmarked version of your revised paper without tracked changes. You should upload this as a separate file labeled 'Manuscript '. If you would like to make changes to your financial disclosure, competing interests statement, or data availability statement, please make these updates within the submission form at the time of resubmission. Guidelines for resubmitting your figure files are available below the reviewer comments at the end of this letter.

We look forward to receiving your revised manuscript.

Kind regards,

Pikka Jokelainen, DVM, MPG, PhD

Academic Editor

Annapaola Rizzoli

Section Editor

Shaden Kamhawi

co-Editor-in-Chief

Paul Brindley

co-Editor-in-Chief

**Additional Editor Comments:**

Dear Authors, thank you for the revised manuscript. One of the reviewers has one additional comment - asking you to expand on in Discussion why attitudes were the least studied domain.

Editorial comments:

Please align the number of decimals used – e.g. in abstract when referring to proportion of studies, none, one and two decimals are currently used. With 16 included articles, decimals seem too much, and readers expect the numbers. Please edit accordingly, e.g., “The studies were conducted in nine LMICs, with Nigeria (37.5%) and Ethiopia (18.75%) accounting for the highest number.” -> “The studies were conducted in nine LMICs, including six studies (38%) in Nigeria and three (19%) in Ethiopia.

Consider the use of ‘only’ when reporting results, as it shows the attitude. I usually recommend to let the numbers do the talking.

Why is the need for targeted interventions to improve KAP among abattoir workers and expanding research to include nearby residents urgent?

Is there a need to use two terms, ‘abattoir’ and ‘slaughterhouse’ (there is also ‘slaughter house’), to one concept? These are described as “essential for meat processing”, which sounds surprising. The animals arrive to these alive. Line 80 – consider omitting word ‘significant’ as it is easily read as statistically strong. Consider description e.g. “risks are exacerbated by inadequate waste management”, which sounds like all of these would have inadequate waste management – possible edit could be ‘can be exacerbated’

Author summary needs editing to ensure it is clear and sufficiently specific. Zoonotic diseases can transmit both ways, from animals to humans and from humans to animals – this is correctly stated in Introduction. “causing millions of cases and deaths annually” is not specific enough - Introduction says billions of cases and millions of deaths, and specifies it is globally.

Par-veterinarian? Often used term is ‘paraveterinarian’. Please check.

Most disease names are not written in capital full letter in English. E.g. ‘toxoplasmosis’ should be with small first letter; and please note that that study tested for antibodies against T. gondii, and that is not the same as testing for toxoplasmosis, the clinical disease. They use “seroprevalence of toxoplasmosis” but more correct would be to state “Toxoplasma gondii seroprevalence”.

Language checking is needed overall. E.g. line 131 full stop lacking at the end of the sentence. The verbs in Table on page 16 should be all in past tense, now there is present tense: “do not wear PPE” and “wash hand” (should be ‘reportedly washed hands’)

**Journal Requirements:**

1) We have noticed that you have uploaded Supporting Information files, but you have not included a list of legends. Please add a full list of legends for your Supporting Information files after the references list.

2) We note that your Data Availability Statement is currently as follows: "All relevant data are within the manuscript and its Supporting Information files.". Please confirm at this time whether or not your submission contains all raw data required to replicate the results of your study. Authors must share the “minimal data set” for their submission. PLOS defines the minimal data set to consist of the data required to replicate all study findings reported in the article, as well as related metadata and methods (https://journals.plos.org/plosone/s/data-availability#loc-minimal-data-set-definition).

3) Please amend your detailed Financial Disclosure statement. This is published with the article. It must therefore be completed in full sentences and contain the exact wording you wish to be published.

4) As required by our policy on Data Availability, please ensure your manuscript or supplementary information includes the following:

**Reviewers' comments:**

Reviewer's Responses to Questions

**Key Review Criteria Required for Acceptance?**

**Methods**

-Are the objectives of the study clearly articulated with a clear testable hypothesis stated?

-Is the study design appropriate to address the stated objectives?

-Is the population clearly described and appropriate for the hypothesis being tested?

-Is the sample size sufficient to ensure adequate power to address the hypothesis being tested?

-Were correct statistical analysis used to support conclusions?

-Are there concerns about ethical or regulatory requirements being met?

Reviewer #2: (No Response)

**Results**

-Does the analysis presented match the analysis plan?

-Are the results clearly and completely presented?

-Are the figures (Tables, Images) of sufficient quality for clarity?

Reviewer #2: (No Response)

**Conclusions**

-Are the conclusions supported by the data presented?

-Are the limitations of analysis clearly described?

-Do the authors discuss how these data can be helpful to advance our understanding of the topic under study?

-Is public health relevance addressed?

Reviewer #2: (No Response)

**Editorial and Data Presentation Modifications?**

Reviewer #2: in discussion section expand on why attitudes were the least studied domain?

**Summary and General Comments**

Reviewer #2: (No Response)

PLOS authors have the option to publish the peer review history of their article (what does this mean? ). If published, this will include your full peer review and any attached files.

**Do you want your identity to be public for this peer review?** For information about this choice, including consent withdrawal, please see our Privacy Policy .

Reviewer #2: No

**Figure resubmission:**
---

## [Editor Report · Decision Letter 2]

10 Jan 2026

PNTD-D-25-00848R2

Scoping review of knowledge, attitude, and practice to zoonotic diseases among abattoir workers and residents in proximity to abattoirs in low-middle income countries

PLOS Neglected Tropical Diseases Dear Dr. Patricia Uche Ogbo,

Thank you for submitting your manuscript to PLOS Neglected Tropical Diseases. After careful consideration, we feel that it has merit but does not fully meet PLOS Neglected Tropical Diseases's publication criteria as it currently stands. Therefore, we invite you to submit a further revised version of the manuscript.

Please submit your revised manuscript by  07/02/2026 If you will need more time than this to complete your revisions, please reply to this message or contact the journal office at plosntds@plos.org. Please include the following items when submitting your revised manuscript:

* A letter that responds to each editorial comment. You should upload this letter as a separate file labeled 'Response to Reviewers '. This file does not need to include responses to any formatting updates and technical items listed in the 'Journal Requirements' section below.

* A marked-up copy of your manuscript that highlights changes made to the original version. You should upload this as a separate file labeled 'Revised Manuscript with Track Changes '.

* An unmarked version of your revised paper without tracked changes. You should upload this as a separate file labeled 'Manuscript '.

We look forward to receiving your revised manuscript.

Kind regards,

Pikka Jokelainen, DVM, MPG, PhD

Academic Editor

Annapaola Rizzoli

Section Editor

Shaden Kamhawi

co-Editor-in-Chief

Paul Brindley

co-Editor-in-Chief

**Additional Editor Comments:**

Editorial comments: All major comments have been considered. Minor language checking and editing is still needed, please see comments below for guidance.

Please check the use of singular vs plural for ‘attitudes’ and ‘practices’ – typically both should be in plural throughout the text.

In Table 1, the first column is indicated to tell Name of first author and year of publication, but it includes ‘et al.’ as well as name of all authors when there was two authors.

Use of capital first letters needs further check. Why currently capital letter for Pleuropneumonia, why not for cysticercosis? Crimean-Congo hemorrhagic fever is written with capital letters for both ‘Crimean’ and ‘Congo’, and with hyphen, by e.g. WHO (Table 1, Table 2).

Toxoplasma gondii needs to be written with capital first letter for genus name, and Italics for both genus and species name, in both text and Table 1.

In Table 2, please check and clarify, these two sentences leave confusion: “292 (97.3%) participants of 300 were found to consume raw meat. 225 (75%) had eaten or consumed raw meat”

Consider and check the use of abbreviations. Are e.g. SH-worker and bTB needed? Currently bTB is explained two times.

In list of included studies, word ’applicable’ is misspelled. Is S/N 4. included? There is no remark for that one. Abbreviation “S/N” is not explained in the document.

American and British spelling are currently mixed (e.g. minimize; behaviour), please check for consistent style.

**Journal Requirements:**

1) We have noticed that you have uploaded Supporting Information files, but you have not included a list of legends. Please add a full list of legends for your Supporting Information files after the references list.

2) As required by our policy on Data Availability, please ensure your manuscript or supplementary information includes the following:

**Reviewers' comments:** 

**Figure resubmission:**
---

## [Editor Report · Decision Letter 3]

26 Feb 2026

Dear Patricia Uche Ogbo,

We are pleased to inform you that your manuscript 'Scoping review of knowledge, attitudes, and practices to zoonotic diseases among abattoir workers and residents in proximity to abattoirs in low-middle income countries' has been provisionally accepted for publication in PLOS Neglected Tropical Diseases.

Best regards,

Pikka Jokelainen, DVM, MPG, PhD

Academic Editor

Annapaola Rizzoli

Section Editor

Shaden Kamhawi

co-Editor-in-Chief

Paul Brindley

co-Editor-in-Chief

---

## [Editor Report · Acceptance letter]

Dear Dr. Ogbo,

We are delighted to inform you that your manuscript, "Scoping review of knowledge, attitudes, and practices to zoonotic diseases among abattoir workers and residents in proximity to abattoirs in low-middle income countries," has been formally accepted for publication in PLOS Neglected Tropical Diseases.

Best regards,

Shaden Kamhawi

co-Editor-in-Chief

Paul Brindley

co-Editor-in-Chief
